# Mental and Physical Health Problems as Conditions of Ex-Prisoner Re-Entry

**DOI:** 10.3390/ijerph18147642

**Published:** 2021-07-18

**Authors:** Anna Pękala-Wojciechowska, Andrzej Kacprzak, Krzysztof Pękala, Marta Chomczyńska, Piotr Chomczyński, Michał Marczak, Remigiusz Kozłowski, Dariusz Timler, Anna Lipert, Agnieszka Ogonowska, Paweł Rasmus

**Affiliations:** 1Department of Clinical Pharmacology, Medical University of Lodz, 90-153 Lodz, Poland; anna.pekala-wojciechowska@umed.lodz.pl; 2Department of Applied Sociology and Social Work, Faculty of Economics and Sociology, University of Lodz, 90-136 Lodz, Poland; andrzej.kacprzak@uni.lodz.pl; 3Department of Medical Psychology, Faculty of Health Sciences, Medical University of Lodz, 90-425 Lodz, Poland; krzysztof.pekala@umed.lodz.pl; 4Department of Forensic Sciences, Stanisław Staszic State University of Applied Sciences in Piła, 64-920 Piła, Poland; mchomczynska@puss.pila.pl or; 5Department of Sociology of Art, Faculty of Economics and Sociology, University of Lodz, 90-136 Lodz, Poland; piotr.chomczynski@uni.lodz.pl; 6Department of Management and Logistics in Healthcare, Medical University of Lodz, 90-131 Lodz, Poland; michal.marczak@umed.lodz.pl (M.M.); agnieszka.ogonowska@stud.umed.lodz.pl (A.O.); 7Center of Security Technologies in Logistics, Faculty of Management, University of Lodz, 90-237 Lodz, Poland; remigiusz.kozlowski@wz.uni.lodz.pl; 8Department of Emergency Medicine and Disaster Medicine, Medical University of Lodz, 92-213 Lodz, Poland; 9Department of Sports Medicine, Medical University of Lodz, 92-213 Lodz, Poland; anna.lipert@umed.lodz.pl

**Keywords:** hostage, social desirability, social isolation/alienation, forensic sciences, health care/public health, social exclusion

## Abstract

The article focuses on a less-discussed issue of social marginalization of people leaving penitentiaries, which is the prevalence of multifaceted health problems experienced by people in this category. It includes poor health status, resulting from, among others, poor housing conditions, harmful or risky lifestyle, and lack of access to medical services. Data from the District Inspectorate of the Prison Service in Lodz, Poland on the health conditions of inmates was accessed. These data were supplemented by qualitative research conducted in 21 juvenile detention centers and 8 prisons across the country, conducting direct observations and In-Depth Interviews (IDI). A total of 198 IDIs were conducted with incarcerated (72) and released (30) juvenile offenders, and incarcerated (68) and released (28) adult offenders. These were complemented by IDIs with experts (50) and Focus Group Interviews (FGIs; 8) with male and female inmates in 4 Polish prisons. The study revealed that mental and physical health is a serious obstacle to social reintegration of ex-prisoners. It is rarely addressed by state institutions. There are strong associations between neglect of health issues in the prison population and increasing social exclusion after leaving prison. As Poland has a restrictive penal policy, former prisoners remain a group with social stigma and little support.

## 1. Introduction

Crime, as one of the most serious social problems, generates direct and indirect costs to the society. Although health-related issues are vital, they are rarely the focus of academic attention. Criminologists mainly focus on health damages suffered by crime victims, including physical harm and long-term psychological disorders such as anxiety, depression or post-traumatic stress disorder [1,2,3,4]. They indicate that these harms, incurred mainly at the individual level, tend to become much broader problems in a synergistic effect, as they can affect the socio-economic situation of different social groups, communities, or even the society as a whole [1,2,3,5,6,7,8,9,10].

In this paper the goal is to provide a criminological perspective on a particular issue related to the topic of public health, i.e., health problems of current and former prisoners as a specific context to their re-entry into the society. Authors argue that as this topic lies at the intersection of health, penal, and social policy, it addresses a particularly serious and fraught manifestation of social inequalities in contemporary societies. Although researchers are divided in analyzing the relationship between social exclusion and health issues [11], they agree that limited access to public health services is an important dimension of social exclusion. Health problems are considered one of the common causes of socioeconomic degradation and, conversely, experiencing social marginalization often contributes to deterioration of health [12,13].

Many former prisoners suffer from mental health issues. A report by The House of Commons [14] reveals a high prevalence among male and female prisoners of such issues as: psychotic disorders (7% and 14%, respectively), anxiety (21% and 32%), depression (33% and 51%), and personality disorders (64% and 50%). Moreover, almost every fifth prisoner has attempted suicide in the past [14,15]. This indicates that in many cases mental health issues were most likely already present before incarceration. It can also be assumed that due to the multidimensional deprivation (social, psychological, sensory), which is one of the main features of total institutions, the prison environment promotes the aggravation of these problems, as well as the emergence of new ones.

The scope of commonly reported physiological issues experienced by prisoners is alarmingly long. S. Grassian and N. Friedman [16] point out that frequently-reported symptoms of deteriorating health condition while in prison include migraine headaches, profound fatigue, gastro-intestinal problems, poor appetite and weight loss, cardiovascular problems (most notably, heart palpitations), genito-urinary problems, diaphoresis, tremulousness, insomnia, joint (esp. back) pains, eyesight deterioration, weakness and lethargy, and aggravation of pre-existing medical problems [17]. S. Shalev [16] argues that these health conditions may in fact reflect psychological stress caused by prison isolation.

Prison, as an acute form of social exclusion [18], creates a number of difficulties in accessing key areas of social life. G. Sykes [19] distinguishes prison disadvantages such as: (a) deprivation of liberty (limitations of contacts with the outside world, freedom of action and movement, manifested by specific architecture of prison facilities: barb wires, pavilions, narrow corridors, separate cells, bars, locks, cameras [4,20,21]; (b) material deprivation (lack of access to consumption services, poor living conditions, e.g., humidity, noise, unpleasant odors, poor lighting, compulsory cell changes, bad food quality [22]; (c) deprivation of heterosexual contacts (which may result in impaired identity, psychosexual development disorders, or—in extreme cases—sexual aggression towards fellow inmates, which often causes psychological and physical damage to the victims [23,24,25]; (d) deprivation of opportunities of independent decision-making (e.g., time-managing or choosing company, which are replaced by a system of prohibitions and obligations and total control exercised by prison staff); and (e) deprivation of a sense of security (due to the constant presence of other inmates, who are generally considered more aggressive than other populations and therefore more dangerous, both physically and psychologically) [22].

American psychologist C. Haney notes: “At the very least, prison is painful, and incarcerated persons often suffer long-term consequences from having been subjected to pain, deprivation, and extremely atypical patterns and norms of living and interacting with others” [4]. Psychological reactions to these afflictions include learned passivity and dependency attitude, hypervigilance accompanied by compulsive interpersonal distrust, psychological distancing and self-alienation, emotional denial which may result in emotional flatness, incorporating exploitative attitudes towards others due to the permeated values of the prison subculture, diminishing self-worth, and post-traumatic stress reactions or “re-traumatization” [4,16,26].

Many studies also show that the percentage of inmates who smoke tobacco, drink alcohol, or take drugs in a hazardous manner by far exceeds the corresponding proportion in the general population [27,28,29,30]. The British Rebalancing Act indicates that 63% of male and 39% of female prisoners can be identified as hazardous/harmful drinkers, compared with 30% and 15%, respectively, in the general population. Approximately 80% prisoners are smokers, which is four times more than the general population. Prevalence of opiate drugs use a year prior to imprisonment is 21% among male and 26% among female prisoners, while the corresponding prevalence among the general population is only 0.2% [31]. Analyses by the American Bureau of Justice Statistics [32], which include over one and a half million prisoners, indicate that persons addicted to psychoactive substances constitute the majority of the population in state and federal prisons (58.5% and 63.3%, respectively). Although there is a clear understanding that substance abuse is one of the major health-risk issues among prison population, there have been only a handful of research projects touching on the subject of substance abuse and addictions among Polish prisoners. M. Kiryluk found that 82% of the surveyed prison doctors estimated that alcohol addiction is more common among prisoners than the rest of the society [33,34] and A. Szymanowska identified that at least 40% of male prisoners has a drinking problem and 5% are addicted to other drugs. Simultaneously, official data only lists the number of inmates who have been registered to therapy. The scale of the problem therefore remains unknown.

Therapeutic interventions are an important tool in the treatment of inmates experiencing alcoholism [35,36]. In 2015 the Supreme Chamber of Control (pl. Najwyższa Izba Kontroli) [37] very critically assessed the availability of therapeutic wards in the Polish prison system, considering that the actual demand was much higher. At that point, 69 therapeutic wards were in operation, while the prison population exceeded 70,000 people. This was evidenced by the long waiting period for admission to a therapeutic ward—as of 30 June 2014, it was over 14 months for prisoners addicted to alcohol and 12 months for those addicted to other narcotic/psychotropic substances [37]. Prison data confirm these difficulties. The percentage of inmates qualifying for these units between 2002 and 2019 has never exceeded 5%. At the end of 2020 (31 December) 2856 adult prisoners of whom 179 women were qualified to therapeutic ward inside corrections and 564 (24 women) to outside facilities. One year earlier (December 31, 2019), 3315 inmates were qualified for therapy, including 1169 for alcohol dependence and 392 for addiction to narcotic drugs or psychotropic substances [38] (Annual Statistical Information for 2019/2020: 32). Moreover, inmates rarely participate in rehabilitation of their own volition [39]. Rather, they are coerced, which results in their reluctant attitudes and lack of commitment [40].

It is estimated that prisoners globally are 5 to 15 times more likely to be infected with HIV or hepatitis C in comparison to the general population [41]. In Poland, in 2017 almost 10% of the incoming prisoners were subjected to HIV antibody testing [42], which indicates that the prison health care personnel are well aware of risky behaviors of some of the inmates outside the rehabilitation facilities. Therefore, it can also be assumed that various health problems self-reported by prisoners are often related to a lack of health-promoting attitudes, manifested in hazardous, harmful, or unhygienic lifestyles outside prison [10].

According to the newest available data provided by prisonstudies.org, as of June 2021, Poland has the highest prison population size and the second highest incarceration rate among the European Union countries (see Figure 1 and Figure 2). Incarceration rate is calculated as a total number of inmates per 100,000 citizens.

Poland’s position at the forefront of the EU countries in this regard is not a novelty. Since the introduction of the new Penal Code in 1997, imprisonment has been the most frequently implemented punishment in the Polish justice system. The number of inmates has remained relatively high, even though crime rate has been steadily falling since 2003 and are at the lowest level since the 1990s (see Figure 3).

Although many prominent Polish specialists have questioned the effectiveness of imprisonment over the past two decades, the abovementioned relationship shows that Poland tends towards a repressive rather than liberal policy [47,48,49,50]. In this model the institution of imprisonment serves rather to isolate and punish offenders than to support their reentry into the society [35,47].

Particularly interesting is the issue of the quality of medical care in prisons and its impact on the lives of inmates after they leave prison. The authors argue that prison facilities, considered institutions of social rehabilitation, may be viewed as an environment in which inmates’ health problems can be addressed to provide favorable conditions for their reentry into society and thus reduce the risk of reoffending. The aim of the article is to demonstrate how ex-prisoners’ physical and mental health issues interfere with their re-entry process.

## 2. Materials and Methods

In order to obtain quantitative data on the health issues faced by the Polish prison population we collected public reports provided annually by the Ministry of Justice and the Prison Service. We also obtained general statistical data from the District Inspectorate of the Prison Service in Lodz, not violating medical confidentiality or accessing inmates’ personal data. Both of these sources provided a very narrow spectrum of information. Moreover, the data collected from single facilities under the District’s management were not suitable for statistical analyzes as they turned out to be rudimentary, undetailed, and unstandardized. Due to these limitations it was decided to focus on the qualitative data.

This article is based on three different projects. The first and most recent of these [51] was granted by the Polish Ministry of Justice and took place between 2018 and 2020 (Grant of the Polish Ministry of Justice, No. DFS-II-7211-169/18/18 titled: “Social determinants of juvenile and adult crime” (project coordinator: Piotr Chomczyński), full report on this research project (in Polish) is available online [51]). In this project, we went through several stages of the consent process. First, the Ministry of Justice, after reviewing the project’s methodology, agreed to analyze and access the data. We then discussed the questions with psychologists and prison management and, after their approval, decided to give juvenile and adult inmates a printed version of the consent text to sign, after providing them with full information about the study objectives, questions, anonymization protocol, purpose, and data use. Because both juvenile and adult offenders were mostly reluctant and suspicious of signing the consent as “official paperwork”, they agreed to give verbal consent at the beginning of the interview recording after being informed of the study details.

A total of 130 individuals participated in the study, of which 90 were juvenile (30) and adult (60) offenders, of which 30 were recidivists. More than half of the offender population was serving sentences in prison (55) and the rest were already at large (35). Juvenile and adult offenders were convicted of various types of crimes (robbery, drug trafficking, theft, homicide, physical assault). Most (28) of the interviews with experts (40) were conducted in both juvenile (correctional) and adult (correctional and detention) facilities. They worked as psychologists, social workers, rehabilitation specialists, professional counselors, educators, and prison staff.

The second project was based on a long-term organizational ethnography between 2008 and 2017. One of the authors (P. Chomczyński) spent a total of 9 months in all of the types of reformatories and juvenile detention centers for boys (17) and girls (4) across Poland. The research was based on open-ended interviews conducted with male (43) and female (29) inmates aged 13 to 21 years. Using overt participant observation, the author participated in all activities performed by the inmates.

The third project took place between 2011 and 2018 and was based on qualitative research methods (Grant of the Polish Ministry of Science and Higher Education, targeted subsidy for young scientists, No. 545/172 titled: “Freedom—an analysis of life strategies and life stories of people leaving prisons. Conditions and barriers to social readaptation”). One of the authors (A. Kacprzak) interviewed (28) former inmates (recruited through NGOs and social welfare institutions) and current prisoners in correctional facilities for adults (8) across Poland. This research was complemented by semi-structured interviews (10) with experts working with former or current prisoners (social workers, psychotherapists, and NGO workers). The study focused on barriers to social inclusion of former prisoners re-entering the society, as well as to desistance from crime.

In the second and third projects, the researchers discussed both their methodology and ethical issues in departmental meetings and went through the appropriate research council in their department where the research objectives, methodological and ethical issues, and data use were discussed and approved. Researchers were required both to inform their interviewees of the details of the study and to obtain verbal consent from them before beginning the interviews. In all three projects the data we obtained both from juvenile and adult offenders was fully anonymized. To ensure the anonymity and confidentiality that are especially needed for sensitive topics, all of the names used in this research are pseudonyms [52,53,54,55]. The length and content of the quotations was edited when necessary to protect our subjects from being identified.

The abovementioned three projects were recently (2019) completed with 8 focus group interviews (FGIs) with male (45) and female (15) prisoners from 4 prisons subordinate to the Lodz Regional Inspectorate of the Prison Service (Grant of the Polish Ministry of Justice, No. DFS-II-7222-33/18 titled: Restart—I start from scratch (project coordinator: Marek Kotlicki, Altermed Foundation); the research results were published (in Polish) in Kacprzak 2021 [56]). The interviews concerned the topic of the living conditions and the quality of services in prison.

The work with juvenile and adult offenders relied on open-ended biographical tools designed to allow our interviewees to make spontaneous statements [52,53,57,58]. To make the relationship between researchers and interviewees more balanced, respondents were invited to develop and co-create interview scenarios by adding their own questions or editing those of the interviewers. This resulted in a richer description than a formal interview tool, and our interviewees became less suspicious. An atmosphere of trust and openness was created to obtain in-depth and profound responses [53,54,58].

Triangulation of our data was particularly important, as the researchers were outsiders and dealing with a sensitive topic. Triangulation of researchers, including outside observers and collaborators, data using a variety of information sources and tools, employing open-ended, semi-structured, and focus interviews, as well as direct observations [59,60]. At all times general data have been used without violating medical confidentiality nor consulting personal data. Informed consent has been obtained from all subjects involved in the study both for IDIs and FGIs.

In all three projects, data analysis was conducted by means of ATLAS TI software (ATLAS.ti Scientific Software Development GmbH, Berlin, Germany). In all of the projects we adopted inductive reasoning, theoretical sampling, and theoretical saturation procedures [60,61,62]. Key categories were inductively generated through open and selective coding of interviews and illustrated with quotes presented in the paper that best reflected the conceptual framework discussed here [61,62,63]. Particular stages of research progress were discussed in department meetings.

## 3. Results

The research results have been divided into two main threads which correspond to the biographical trajectory of persons sentenced to prison: the period of stay in prison and the period after their release. The first part focuses on the assessment of the quality of health care faced by prisoners in the Polish penitentiary system. The second refers to the impact of inadequate support in addressing their health problems upon the re-entry process.

### 3.1. Health Care during Imprisonment

Addiction to alcohol or drugs is the most common health problem among prisoners. Surveyed prison therapists estimate that alcoholism affects between 25% and 70% of the prison population.

Interviewee (I): I actually came here [to the therapeutic ward] under duress, because I did not agree to the therapy myself, because I did not want to. The psychologist referred me to the penitentiary court, they stated in article 117 that I must go to treatment, because I have a problem, I’m addicted, right? I denied it (…) I did nothing, I didn’t participate [in therapy]. I fucked this whole thing. I stood by my own, that I have no problem with drugs, that I can handle it. (…) They are sick, they want to put some disease in my head. Which I don’t really have. (…) So, when I was here on the ward, I endured drugs. My brother used to visit me from freedom to see me, he brought me heroin, amphetamines, needles, syringes. <adult male, recidivist>

Psychoactive substances abuse also affects juveniles from correctional facilities. Moreover, the ESPAD research [64] shows that in contemporary Poland the difference in harmful use of alcohol and drugs between boys and girls is blurring. The problem is visible, and concerns not only alcohol and drugs, but also so-called power-ups. Simultaneously, educators from youth shelters argue that the offer of addiction treatment is rather limited.

I: When they are in the transition room, they go through this withdrawal period and then they end up in the [support] group and that’s it. And later, if an addiction is diagnosed (…) he may go to a correctional facility in Białystok, and that is that. There are simply no specialists at the shelter who would do it properly. <counselor, juvenile facility>

The problem of inadequate medical care being offered in Polish prisons, however, is much wider. Each of the eight focus groups demonstrates that prisoners describe prison health care as extremely poor. Mainly, due to the lack of accessibility to these services (related to the presence of a doctor in the facility only on selected days of the week), complemented by the apparent neglect of health issues by prison staff. One inmate explains using his own example:

I: You cannot even ask for a visit to the doctor, because the waiting time is 3 weeks, for an antibiotic—4 months. So, either you will manage to heal yourself or—as I had it—the disease will spread to the whole body, because I have a [skin disease] and I needed such a specific medicine (…) and I got to the doctor after 4 months. It was during this time that it developed all over my skin, everywhere. <adult male, 1st time prisoner>

Prison medical care was rated very poorly by the female offenders participating in the FGIs. There was a consensus on this point, with negative opinions appearing regardless of age, prison tenure, or even current health status.

I: Doctors disregard us (…) Healthcare here is at a zero level. Polopyrine here is [a cure] for everything. The girl from my cell got it, for example, for inflammation of the peritoneum. <adult female, 1st time prisoner>

In addition to long waiting periods for medical appointments and a lack of basic medications that should be available in case of an emergency, commonly reported problems with prison health care include staff downplaying health problems or suspecting inmates of simulating illnesses.

I: It’s horrific here… You can die here. When I was in [Name of the prison], a girl arrived (…) from this prison. And she had in her papers that she simulated a heart defect, because that was what the local doctors said, so they didn’t accept her there either. And she died on the same night she arrived. <adult female, 1st time prisoner>

The prison staff’s distrust can be, to some extent, explained by the fact that some of the inmates simulate sicknesses in order to get preferential treatment or gain access to medications.

Researcher (R): And what is your favorite way of spending your time in prison?

Interviewee (I): I used to go to a psychiatrist to prescribe me sleeping pills. Because my thoughts are so weird. [The doctor] didn’t ask about it, he did not research, he just would rewrite the prescription. Sometimes I collected these [pills] from others so I could sleep all the time. <adult male, recidivist>

This tendency, however, works to disadvantage of those prisoners who suffer from actual ailments, as exemplified below:

I: They [inmates from a hostile subculture group] beat me up so much that they took me unconscious to the [prison] hospital. (…) I will never forget this. They did something to my leg, it started to crush me (…) but when I wanted to go to the doctor they said I was simulating, because the leg is not swollen. However the pain was horrendous. I could not put my foot gently or even rest without pain. When I finally went to the doctor I had a fever of 40 degrees. <adult male, recidivist>

Our data shows that while it is widely believed that mental health problems are risk factors for recidivism, it may not be adequately addressed by the Prison Service—as evidenced by the following statement from one of our Prison Service informants:

I: Especially mental health protection is a big problem, because someone calculated up there that the cost was 12–13 thousand [PLN]. The salary of a psychiatrist is quite a lot and they dismissed a psychiatrist treating prisoners. They offered him a lump sum and he refused as the rates were ridiculous. And the problem of the situation of people with disorders leaving prisons emerged. They then go outside and pose a threat to themselves and others. <Prison Service informer>

Where the economic costs of prison health services are low, neglect of various kinds is likely to occur. This, in turn, may result in (further) social marginalization of inmates after they leave prisons. To the extent that prisons tend to downplay mental health problems and do not address them because such services generate ‘unnecessary’ costs, one can argue that such policies may be a contributing factor to higher recidivism rates. However, as our observations from juvenile prisons show, the opposite risk also exists, manifested in ‘over-diagnosing’. An example is provided by one of the psychotherapists interviewed:

I: Boys very often come to the facility with a diagnosis of a personality disorder, while the literature says that personality is formed until the age of 18–19, so a boy may reveal features only, this personality may only develop, but it irritates me when I see a diagnosis of a borderline personality disorder in a 15-year-old. Often, psychiatrists write a personality disorder, yet they combine it with ADHD and the matter is over the top. (…) And they usually already have labels attached to their documents. <psychologist, juvenile facility>

Particularly drastic manifestations of mental health problems experienced by the residents of correctional facilities are their acts of self-injury. However, our research shows that the motivations for inflicting self-harm are different in the prison population and among juveniles in correctional facilities. According to our study, among juveniles, self-harm is more emotionally determined; it is a harbinger of mental problems (nothing happens to them—they are not placed in therapeutic facilities, etc.) [65].

R: Why did you cut your hands?

I: Because I had such a period, because of various problems… For example, my mother turned against me when she was with her boyfriend, it was also because of this. <female, juvenile facility>

On the other hand, the observation of juvenile behavior shows that self-harm is also repeated after more popular peers with a higher position in the group, which indicates non-emotional motivation [65].

R: I noticed that this [cutting hands] is probably such a fashion. One day, only one girl has her arms cut, another one it’s several…

I: Because some [girls] are so cool and all. I know this one… she thinks it’s cool and that she likes to cut her hands, because it looks fancy. And she had such a cut, ugly, I wouldn’t be able to cut it this way to make it look asymmetrical somehow. <psychologist, juvenile facility>

Self-injury among juveniles is largely dictated by emotions and correlated with the individual’s psycho-emotional condition. Interestingly, this type of self-harm is more common among girls than boys. Boys tend to prefer body tattooing, perceived as an image associated with masculinity [65,66].

In the case of adult offenders, self-injury is more instrumental and involves a secondary adaptation [67] calculated for planned gain (cold calculation). Adults know that for self-harming they can obtain tangible benefits, e.g., transfer to a therapeutic institution or to a hospital ward where ’one can serve time better’ (thus they are associated with simulating an illness or an immediate threat to health), exerting pressure on staff, educators, or prison management. At other times, self-harm is a method of impressing fellow inmates, which resembles the strategy of imitation among juveniles. Among adults, however, self-aggression as an emotional response occurs less frequently. If it does, it is likely to be serious in nature, even aiming to take one’s own life. Importantly, our research suggests that such acts mainly affect inmates with no prior experience of prison or experienced offenders with particularly long sentences.

I: I told myself that I would not be held responsible and that I would not be in prison for so many years… I would cut my wrists as soon as I step into detention. After they locked me up, I slashed my veins on the second day. I was lying in this bed, I took out the artery (…) About three hours I was bleeding out (…) I was getting cold, I was getting hot, I started to sweat (…) I reached out for cigarettes and lost consciousness. <adult male, recidivist>

In contrast, some interviewees who were incarcerated while in very poor health condition tend to present a somewhat different perspective, identifying prison as a good setting to return to a better health condition and, eventually, to the risky or harmful lifestyle led prior to sentencing. This strategy applies to a large group of prisoners addicted to alcohol or hard drugs. One interviewee recalls:

I: When I got imprisoned (…) I had so many eschars [caused by self-injecting with heroin] that they had nowhere to insert the needle… legs and hands (…) They had to put cannulas in my neck (…) [With three other people] I took a kilo of mephedrone and almost a kilo of heroin over the span of three months. It scares me terribly and I thank God that I am actually in prison. <adult male, recidivist>

After the recovery period, he returned to drug use in prison and did not undertake any therapeutic work. He was not interested in participating in any rehabilitation activities. From a rehabilitation perspective, such prisoners appear to have a relatively poor prognosis for the future, as a large proportion of them are primarily interested in repairing their condition and then ‘waiting out’ their sentence without engaging in restorative activities.

### 3.2. Health Issues as Barriers to Re-Entry

Although most of the interviewees evaluate prison health care as poor, insufficient, and inadequate to their actual needs, there can be an argument made that incarceration provides an opportunity to address some of the health issues experienced by the prisoners. Despite the hostility of the prison environment, for inmates who lack social and economic resources and experience homelessness outside prison, the incarceration period may paradoxically create a chance to improve their health. For those who are addicted to psychoactive substances, the stage of incarceration promotes staying sober through a strict limitation of access to alcohol and drugs consumption. Upon release, however, when these restrictions disappear, many inmates relapse [32,68].

I: The fact that [a prisoner] was serving a sentence does not mean that he does not come back [to the therapy ward]. He does. I am working in a substitutive therapy unit, i.e., where an addicted patient takes methadone. He comes back anyway. Despite our efforts. So my activity is mainly limited to helping him find himself in the society, again and again. <psychotherapist, prison therapy ward>

The same pattern applies to juvenile offenders:

I: One thing is worth noting: the fact that they do not have access to drugs in the facility does not mean that they are already clean. They declare that as soon as they leave, they will “pull in their nose”. <psychotherapist, juvenile facility>

Many former prisoners, especially repeat offenders and those with particularly long sentences, do not believe it is possible to lead a “normal life” outside of prison. One former prisoner recalls:

I: I was waiting for this day of departure, you know it! (…) And the closer to this exit, the more I was overwhelmed … Well, to put it so brutally, I had terrible poops. I didn’t want this previous life anymore, but I was afraid. <adult male, ex-prisoner, recidivist>

The first few weeks after leaving prison are widely regarded by the experts interviewed as a critical period for the future of ex-prisoners. It is often at this stage that former inmates return to their former environments in search of support. This tendency, however, greatly increases the risk of returning to the activities that contributed to their criminal activity in the first place.

I: Most often, these people go to the same places, to the same people (…) with their old habits (…) while in statistics about 70% of crimes [are those committed] under the influence of alcohol. (…) Human pro-health behaviors outside prison are absolutely crucial if you don’t want to come back here. But when funds run out, you have to find a way to get them, and under the influence of alcohol it is easier not to worry that the crime will be exposed. This is most often the case of robberies and the circle is closed. <counselor, adult male prison facility>

Paradoxically, however, the interviews conducted show that leaving the prison is the stage in which former inmates are left to rely only on themselves. They receive a small amount of material support, sometimes symbolic financial support for traveling to the place of stay, and a list of institutions to which they can apply for help. One of the inmates interviewed, who has a severe disability (his leg was amputated while in prison due to gangrene caused by peripheral artery disease), describes his experiences in this matter:

I: (…) so I‘m leaving prison. I lost my apartment [while serving a 15-year sentence], I don’t have anywhere to go. I got, I think, 150 PLN, which is actually a maximum for what they give you (…) I got some clothes there, but you can probably imagine their quality (…) So where will I sleep? I spend the first night at my friend’s, the second at another’s apartment. And that’s it. That’s the help you receive. And now, how do you survive? Even if you tried I don’t know how hard, you won’t be able to live on. <adult male, ex-prisoner, recidivist>

Another interviewee recalls a situation in which a fellow inmate leaving prison is not the focus of the Prison Service or support institutions, despite his poor mental state and possibly also lack of support from loved ones:

I: There was a boy who came out of here, mentally ill, came out of here and stood in the gate and didn’t know what to do with himself. Then he started walking across the field, into the trees and back to the gate. He didn’t know where to go, what to do. The ambulance had to come, and they asked what was wrong with him. They took him to the counselor, and the counselor just said that the boy was mentally ill. So they took him to the hospital. <adult male, recidivist>

One can question why a person in such poor psychological condition served a prison sentence in the first place. The interviewed experts almost unanimously indicate that mental problems experienced by former prisoners are often severe and need to be addressed with professional and sometimes immediate and intense support.

I: Prisoners after larger sentences, such as few years old, I think they are more closed and withdrawn (…) They are often so damaged and so personality disturbed after being released from prison that they need a longer, deeper, and more intense work on themselves. <prison psychologist>

Unfortunately, however, after leaving the facility, ex-prisoners once again collide with the limitations of the public health service.

I: Access to free psychological care is crucial (…) so that in a difficult, mutagenic situation of relapse, this person could quickly contact a specialist who would help him and prevent a mishap (…) or relapse. As far as I know, the waiting period for a psychological consultation under the National Health Fund is about 12 months and 2–3 months to see a psychiatrist. (…) Some of them can afford private medical advice but most are unable to pay for such benefits, it’s difficult to find a job and cope with the life outside prison, and even more so in paying for medical advice, which is not the cheapest. <prison psychologist>

The interviewees emphasized that a particularly serious problem at the stage of re-entry is the ex-inmates’ habituation to the realities of prison life due to their long-term institutionalization. One of the interviewed therapists refers to this phenomenon as “the prison sickness”. The prison institution triggers psychological changes. A person who had spent some time in isolation from the outside world, being under the 24-h supervision of prison staff, learns to be passive. For this reason, many are helpless when released from prison and may even begin to miss living in a prison. The aforementioned therapist compares this emotional state to addiction.

I: Let’s be clear about this. Depriving someone of physical freedom is violence. (…) A person who experiences violence learns to miss it. These people miss this system. These mechanisms are like hypocrisy in any addiction. A man will subconsciously do everything to get there once again. <therapist>

From the criminological point of view, two main life paths of people leaving prison can be distinguished, i.e., continuation of criminal career, and decriminalization, i.e., its suspension or permanent termination. However, desisting from crime requires a number of difficult endeavors, such as getting a job, finding a place to live, receiving income, securing social needs, and addressing health issues. It is a mentally aggravating process, which is additionally reinforced by the social aversion to people with a criminal past. This is especially visible in the labor market. The stigma of a prison past makes employers reluctant to hire former prisoners. In conjunction with the barriers experienced by some of them in other spheres of social life, this triggers (further) marginalization. This phenomenon is especially acute for former prisoners experiencing serious health issues, e.g., chronic diseases that prevent them from finding employment. Particularly striking in this context are the experiences of the interviewed prisoners with disabilities, who become subjects of cross-stigmatization, on one hand due to their disability, and to their past imprisonment on the other.

I: Why can’t I find a job? I tried, I really tried (…) but now I’m constantly unemployed. (…) when I went to the a commission [for assessing the degree of disability], they told me “You speak rather well, you hear a little, you are capable, you can go to work”. Really? I am? So why doesn’t anyone want to hire me? <adult male, ex-prisoner>

The above quote brings to mind the previously-discussed tendency to downplay the ailments of prisoners-patients by prison staff. Given the limited access to public health services both in prison and in the outside world, it may be questioned to what extent public institutions are set to provide support to those who cannot afford private medical visits. The issues described in this section—although briefly—indicate that both experts and the ex-prisoners have a rather critical opinion on this subject.

## 4. Discussion

Modern societies encapsulate different crime prevention strategies in a variety of punitive policy models, ranging from mild and liberal, through moderate and restrained, to restrictive and punitive. The direction taken by the public authority is evidenced, among other things, by the frequency with which the most onerous punishments are used. To such undoubtedly belongs the punishment of imprisonment through deprivation of liberty. Among supporters of its widespread use, imprisonment may be considered the only effective means of rehabilitation of offenders. Its critics, on the other hand, point out that it is not possible to socialize anyone through their social isolation.

The issue of the frequently-reported lack of adequate healthcare during imprisonment is a distinct manifestation of the tendency to pursue a punitive and marginalizing, rather than liberal and integrative, model of punishment. This tendency is at odds with the duties of the Prison Service. Healthcare for prisoners is one of its basic and particularly important tasks, which results from the obligations of the state towards persons completely placed under its supervision [69]. Health problems experienced by (ex-)prisoners at the individual level interact with systemic obstacles: inaccessibility to addiction therapies, lack of medical support for prisoners suffering from chronic diseases, or cutting finance for psychological treatment. In prisons, these are accompanied by unhealthy living conditions (e.g., lack of daylight, private space, poor infrastructure) [70,71]. Outside prisons, they overlap with other dimensions of social exclusion: homelessness, lack of income and economic resources, deficiency of formal and informal sources of support, or unemployment resulting in a lack of health insurance and access to medical services.

A week before leaving prison, prisoners usually feel euphoria and eagerly await the day on which they will be released. This psychological state, referred to as *gate fever* [9,72], rapidly changes after a week outside prison and is replaced by depression and a feelings of alienation [73]. The more than 80-year-old concept of *prisonization* [74] is of particular importance in this respect. It is argued that the more time an individual spends in the prison setting, the more accustomed he or she becomes to its restrictions and limitations. Thus, leaving prison may cause decline in self-esteem, distress, anxiety, and depression [4,75,76]. Such psychological reactions over re-entering the society are, in multiple cases, concerned with the decision to undertake a different, more healthy and less risky lifestyle outside prison. Frequently, however, such plans appear far from the reality once outside prison. It is during this period that the risk of suicide greatly increases. According to Revolving Doors Agency, men and women leaving prisons are 18 times more likely to commit suicide than the general population [15]. At the same time, ex-prisoners who misused substances prior to imprisonment are at particularly high risk of overdosing [14].

Addicted persons experience difficulty adjusting to life in the community after leaving prison. They are less likely to find proper job, more likely to be homeless, and more likely to return to prison [68]. This relationship has been confirmed by the US Bureau of Justice Statistics, among others [32]. A number of studies also indicate that ex-prisoners who do not find support in their family environment experience particularly great difficulties. Because they cannot rely on loved ones for emotional support, they experience more tension in situations of failure. Ultimately, they often manifest symptoms of psychopathology, anxiety, and depression [77], are more likely to return to harmful drinking or drug use [78], and eventually return to prison [79,80].

Most criminologists agree that social exclusion is one of the factors bearing particular risk of recidivism. At the same time, numerous studies show that the process of social degradation affects a significant number, if not the majority, of people who leave prison after serving their sentences [9,27,81,82,83,84,85,86,87]. It is in society’s interest to create conditions in which deprivation of liberty contributes not to their re-criminalization, but to their adaptation to life in society (not only in a pedagogical and behavioral sense, but also in the economic and social sense) [88]. While the quality of medical services in prisons remains mediocre or poor, the experience of solitary confinement contributes to the (further) health (and socio-economic) degradation of inmates [14]. Pre-existing health problems are exacerbated or combined with the new ones as a result of experienced multidimensional deprivation experienced.

The relationship between medicine and imprisonment as a method of responding to crime has a long history and goes back to the very foundation of the idea of rehabilitating offenders. This is evidenced by the presence of medical terminology that is implemented within the model, such as “treatment” or “corrections” [89,90]. T. Ward and S. Maruna [90] argue that over the years this term has gained somewhat of a pejorative connotation in academic discourse, eventually becoming a “dirty word”. The model of offender rehabilitation has been criticized for its pedagogical, normative overtones, as being based on a conception of crime as a social “aberration” and thus the perception of offenders (“criminals”) as “social misfits” whose aberrant behavior can be modified to conform to society’s norms—in other words, “cured” [81].

The concept of offenders’ social rehabilitation has a vertical top-down structure. It requires external (professional) intervention in the process of “change”, which is based on a negatively oriented axionormative diagnosis. Because it tends to focus on and generalize inmates’ “deficits” as causes of their criminal behavior, this approach often fails to recognize their individual resources and potentials [91,92]. Corrective actions are mostly standardized and of limited effectiveness exacerbating the social exclusion of former prisoners.

## 5. Conclusions

Our research proves that prison health care faces many problems, mainly due to the lack of accessibility to services and neglect of health issues by prison staff. Prisoners also suffer from mental health problems that are risk factors leading to recidivism and put social integration process in danger after release from prison. Institutional neglect of health issues resulting in social marginalization of inmates increases the risk of their re-entry to prison.

People addicted to drugs and/or alcohol are overrepresented among the population of ex-prisoners’. When released from prison, many of them do not undertake any work on themselves and are not interested in participating in any rehabilitation activities. Because of this, ex-prisoners are especially vulnerable for social exclusion and stigmatization.

Our research also shows that self-harm is a serious problem both among adult and juvenile offenders. Juveniles are more emotionally determined to self-harm, whereas adult offenders present more instrumental and calculated approaches focused on obtaining tangible benefits, e.g., transfer to a therapeutic institution or to a hospital ward where they will receive better treatment. Prison health care needs reforms that would increase institutional help effectiveness to prevent ex-prisoners from social marginalization and further prison re-entry.

## Figures and Tables

**Figure 1 ijerph-18-07642-f001:**
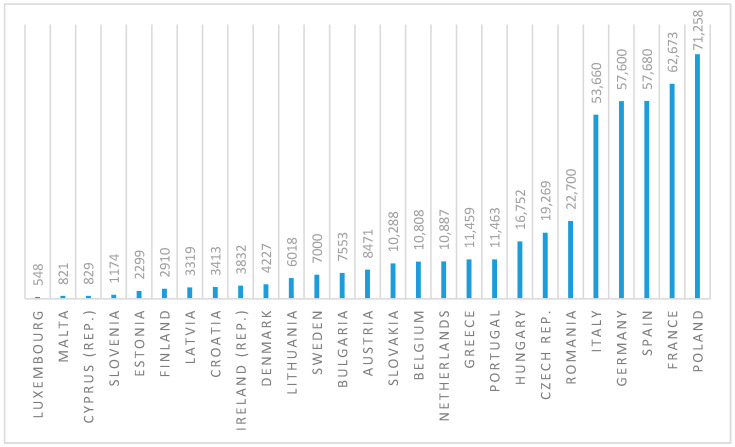
Number of inmates in the European Union countries (June 2021). Source: Authors’ own study based on data retrieved from: https://www.prisonstudies.org/map/europe (accessed on 23 June 2021) [43].

**Figure 2 ijerph-18-07642-f002:**
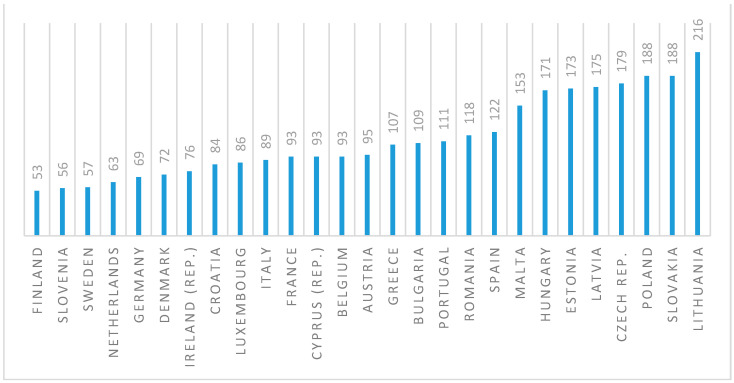
Incarceration rate in the European Union countries (June 2021). Source: Authors’ own study based on data retrieved from: https://www.prisonstudies.org/map/europe (accessed on 23 June 2021) [43].

**Figure 3 ijerph-18-07642-f003:**
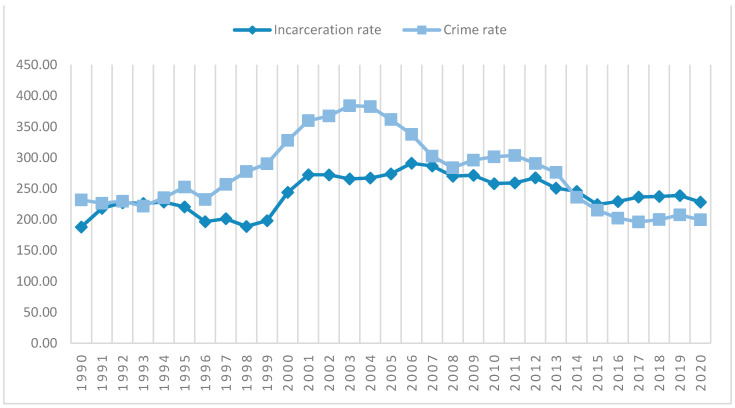
Incarceration rate *versus* crime rate in Poland (1990–2020). For the purpose of visual presentation in Figure 3, the crime rate was calculated as a total number of established criminal cases per 10,000 citizens. Source: Authors’ own study based on data retrieved from: https://www.sw.gov.pl/strona/statystyka-roczna, https://stat.gov.pl/en/, and https://statystyka.policja.pl/ (accessed on 23 June 2021) [44,45,46].

## Data Availability

Non-digital data supporting this study are curated at Medical University of Lodz.

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
