# Peer review of "Mental and Physical Health Problems as Conditions of Ex-Prisoner Re-Entry"

_ijerph, 2021, doi:10.3390/ijerph18147642_

Round 1

Reviewer 1 Report

Please provide more details about the decision to waive the study from IRB ethical review, given that the study involved prisoners and minors, sensitive patient information collected from psychiatrists, well as questions edited/created by participants. 

“this is not a particular concern for the Polish Prison Service” This sentence may be interpreted differently with what the authors meant. Is there a safer phrase that “concern”? An organization’s concern of a problem would be difficulty to conclude, even if such problem is inadequately addressed. 

Health issues and re-entry have been well discussed. The linkage between the two needs to be further established if this is what study aims to conclude.

The casual relation between mental/physical problems and re-extra, as indicated in the title, does not seem to be quite apparent from the results and discussions. 

Author Response

Dear Sir/Madam,

        We wish to express our appreciation for the comments and suggestions for our manuscript entitled “Risk factors of early adolescence in the criminal career of Polish offenders in the light of life course theory”. We have carefully revised the manuscript taking into consideration all the comments.

  • Please provide more details about the decision to waive the study from IRB ethical review, given that the study involved prisoners and minors, sensitive patient information collected from psychiatrists, well as questions edited/created by participants.

We have made the ethical side of the study described in a more detailed way. It is visible in the materials and methods section.

  • “this is not a particular concern for the Polish Prison Service” This sentence may be interpreted differently with what the authors meant. Is there a safer phrase that “concern”? An organization’s concern of a problem would be difficulty to conclude, even if such problem is inadequately addressed. 

We have changed it to: “…it is not adequately addressed…”

  • Health issues and re-entry have been well discussed. The linkage between the two needs to be further established if this is what study aims to conclude.

We have deepened the analysis in this section and provided more data that supports the discussion and conclusions.

  • The casual relation between mental/physical problems and re-extra, as indicated in the title, does not seem to be quite apparent from the results and discussions. 

We developed the physical and psychological issues section in the results section - 3.1. and 3.2. We attempted to make the relation more apparent by strengthening the above mentioned linkage between results and discussion.

Once again we would like to sincerely thank You for a very comprehensive, insightful and crediting review. All of the above comments are highly valuable for the comprehensiveness of the paper and our own scientific development.

Kind Regards,  Dariusz Timler MD PhD, Assoc. Prof.

Reviewer 2 Report

This is an interesting article which highlights issues around prison health care in Poland, where it seems such research is not commonplace. It also explores healthcare issues after release.

It is based on thorough and sound fieldwork (although the first time the abbreviation 'IDI' is used it would be helpful to set out what that is.) The characteristics of the participants could, however, be clearer: this could perhaps be shown in a table form.

It is well-written, although in some sections the use of language could flow more smoothly (and it needs proof-reading throughout). Please check the style guidance for the journal for the use of quotations from interviews, because at the moment these seem to form part of the text and, if possible, it would be good to highlight them more visibly.

The conclusions would benefit from being strengthened, so the recommendations are clearer to the reader

Author Response

Dear Sir/Madam,

        We wish to express our appreciation for the comments and suggestions for our manuscript entitled “Risk factors of early adolescence in the criminal career of Polish offenders in the light of life course theory”. We have carefully revised the manuscript taking into consideration all the comments.

  • This is an interesting article which highlights issues around prison health care in Poland, where it seems such research is not commonplace. It also explores healthcare issues after release.

Thank you.

  • It is based on thorough and sound fieldwork (although the first time the abbreviation 'IDI' is used it would be helpful to set out what that is.) The characteristics of the participants could, however, be clearer: this could perhaps be shown in a table form.

Thank you. We changed the abbreviations to the full names. As far as characteristics of the participants are concerned, our research did not primarily focus on health issues – these “appeared” independently in the process of gathering data in all of the research projects. Thus, we have reanalyzed the material in this respect. Unfortunately, we do not have a possibility to compile the characteristics of our interlocutors into a table as our data on health issues and their treatment is unstandardized.

  • It is well-written, although in some sections the use of language could flow more smoothly (and it needs proof-reading throughout). Please check the style guidance for the journal for the use of quotations from interviews, because at the moment these seem to form part of the text and, if possible, it would be good to highlight them more visibly.

Thank you. We proof-read the article and corrected all possible mistakes. Also, we used different editorial style in order to make quotations more visible.

We have deepened the analysis in the Results section (3.1 and 3.2) in order to make the relation between (ex-)prisoners’ health issues and their re-entry more apparent and therefore strengthen the conclusions as resulting from this linkage.

Once again we would like to sincerely thank you for a very comprehensive, insightful and in many places accurate review. All of the above comments are highly valuable for the comprehensiveness of the paper and our own scientific development.

Kind Regards,

Dariusz Timler Assoc. Prof, MD, PhD

Reviewer 3 Report

The article is interesting, however, there are significant editorial errors and the following issues should be clarified (see my comments).

General comments

1#Introduction -  The authors conducted research in Poland. I miss information on Polish prisons. It suggests adding current problems, statistical data - it will make the article more interesting for people from outside Poland.

2# "Institutional Review Board Statement: Ethical review and approval were waived for this study, due to the specificity ofther study and the lack of any effect on the subject"  Despite this declaration, it needs clarification whether the patients gave informed consent to participate in the study and the use of data / words in the study . If so, please put it in the methods section.

3#Results - I propose to add a table or diagram with the collective problems of prisoners. The results are presented in a way that would make reception difficult. There is a lack of clear results  and statistical analysis.

4#Conclusions - In my opinion, the results do not support the conclusions. There is a lack of clear figures and statistical analysis to support the conclusions. This main remark connects with comments no # 3.

5#Quotes - incorrect version of the quotes in the text. Authors are misusing the version reference number and page numbers. You see specific comments. This should be corrected throughout the text.

6# References - incorrect description of many items. The authors should check all the items in the references description.

" In the text, reference numbers should be placed in square brackets [ ], and placed before the punctuation; for example [1], [1–3] or [1,3]. For embedded citations in the text with pagination, use both parentheses and brackets to indicate the reference number and page numbers; for example [5] (p. 10). or [6] (pp. 101–105).

The reference list should include the full title, as recommended by the ACS style guide. Style files for Endnote and Zotero are available.

References should be described as follows, depending on the type of work:

 Journal Articles:
1. Author 1, A.B.; Author 2, C.D. Title of the article. Abbreviated Journal Name Year, Volume, page range."

https://www.mdpi.com/journal/ijerph/instructions

Specific comments

Abstract

L30-31 - " We accessed data from the District Inspectorate of the Prison Service in Lodz, Poland on the health condition of inmate" Who agreed to this? Were prisoners informed that their data could be used for the study?

L33-" IDIs"- expand the shortcut. You are using it for the first time.

L35 - "FGI" - expand the shortcut. You are using it for the first time.

Key words:

L 41 - All "key words"  are not according with The Medical Subject Headings (MeSH).

Introduction

L43 - delete "Literature overview" - it isn't according with mpdi template

L48 - change "[1](p. 226), [2](pp. 4-8), [3](pp. 177-199), [4](p. 11)"  to "[1-4].

"For embedded citations in the text with pagination, use both parentheses and brackets to indicate the reference number and page numbers; for example [5] (p. 10). or [6] (pp. 101–105)."

In my opinion you don"t citations and you should will use only numbers placed in square brackets.

L51-52 -"[ 1](pp. 219-277), [2](p. 519), [3](pp.  200-204), [5](pp. 32-50),  [6](p. 38-50),  [7](pp.  82.-87), [8](pp.  11-13),  [9](pp.  37-5244), [10](pp. 365-370)" - See notes above.

This should be corrected throughout the text.

L109 -L110 " Many  studies  also  show  that  the  percentage  of  inmates  who  smoke  tobacco,  drink alcohol or take drugs in a hazardous manner by far exceeds the corresponding proportion the general population. " - I can't see  quotes to such a bold thesis.

L123 - " social rehabilitation" - why italics was used?

L125 - suggests adding a clear research aim.

Materials and Methods

L126-170 I miss more information about collecting data, combining it and conducting analyses. Who has consented to the analysis and access to the data. Have the inmates been informed that their data will be used? I suggest expanding this part.

L127 - "This article is based on three different projects." I lack information about the ethical consent for each project or the written consent that the respondents have filled in to participate in it.

L145 - "through"- why italics was used?

Results

L173-191 - This passage is more suited to the introduction than the results. You quote someone else's achievement, not your own.

L177 - " [pl. Najwyższa Izba Kontroli]" change into "(pl. Najwyższa Izba Kontroli)

L195 - "[IN THERAPY]" change into "[In Therapy]"

L225 - "zero level" - why an underscore was used?

L232 - "[NAME OF THE PRISON]" change into "[Name Of The Prison]"

L356-L57 - "(BS6-Bogusław / 60)" - I understand that "Bogusław" is a name. I don't understand what this parenthesis means in the text. What was it placed for?

Discussion

L398 - change  "[501" to "[50]"

L421-422 - "Tony Ward and Shadd Maruna" - you should not use the first names of the authors in the text

L451-459 - Should be used, initials used, not full names and surnames.

Acknowledgments

L470 -"243"

L471 - "244"

L473 - "246"

are these mistakes?

L473 -" Operational Programme Knowledge Education 246 Developmen" - add dot.

Author Response

Dear Sir/Madam,

        We wish to express our appreciation for the comments and suggestions for our manuscript entitled “Risk factors of early adolescence in the criminal career of Polish offenders in the light of life course theory”. We have carefully revised the manuscript taking into consideration all the comments.

General comments

  • 1#Introduction -  The authors conducted research in Poland. I miss information on Polish prisons. It suggests adding current problems, statistical data - it will make the article more interesting for people from outside Poland.

We added current statistical data concerning Polish prisons. They are visible in Introduction section.

  • 2# "Institutional Review Board Statement: Ethical review and approval were waived for this study, due to the specificity ofther study and the lack of any effect on the subject"  Despite this declaration, it needs clarification whether the patients gave informed consent to participate in the study and the use of data / words in the study . If so, please put it in the methods section.

We have made the ethical side of the study described in a more detailed way. It is visible in the materials and methods section.

  • 3#Results - I propose to add a table or diagram with the collective problems of prisoners. The results are presented in a way that would make reception difficult. There is a lack of clear results  and statistical analysis.

In qualitative study like ours it is unlikely to make a separate quantitative analysis. It is a great suggestion for our future work, for instance to use more data and apply quantitative methods (i.e., text mining) to explore the patterns. Due to the fact that all 3 projects discussed in this paper were unique in some ways, we strongly feel that quantitative techniques would not be justified. If we cite official statistics there are very few information covered, they are very general (national ones) and not standardized (from particular areas of Poland). They do not reflect the reality, because uninvolved diagnostics and lack of availability of medical services.

  • 4#Conclusions - In my opinion, the results do not support the conclusions. There is a lack of clear figures and statistical analysis to support the conclusions. This main remark connects with comments no # 3.

Applying qualitative approach, we focused on delivering in-depth and exploratory insight in mental and physical health problems of prisoners. Our aim was to understand nuances lying under this particular health care. Despite statistical representatives we employed theoretical saturation and theoretical sampling to develop better understanding of this complicated subject.

  • 5#Quotes - incorrect version of the quotes in the text. Authors are misusing the version reference number and page numbers. You see specific comments. This should be corrected throughout the text.

We have corrected this mistake.

  • 6# References - incorrect description of many items. The authors should check all the items in the references description.

We have corrected those mistakes.

Specific comments

Abstract

  • L30-31 - " We accessed data from the District Inspectorate of the Prison Service in Lodz, Poland on the health condition of inmate" Who agreed to this? Were prisoners informed that their data could be used for the study?

At all times general data have been used by us, without violating medical confidentiality nor consulting personal data. Informed consent has been obtained from all subjects involved in the study. We made the description of this procedure more detailed.

  • L33-" IDIs"- expand the shortcut. You are using it for the first time.

We changed the abbreviations to the full names.

  • L35 - "FGI" - expand the shortcut. You are using it for the first time.

We changed the abbreviations to the full names.

Key words:

  • L 41 - All "key words" are not according with The Medical Subject Headings (MeSH).

We have corrected this mistake and changed key words.

Introduction

  • L43 - delete "Literature overview" - it isn't according with mpdi template

We have corrected this mistake and changed it to Introduction.

  • In my opinion you don"t citations and you should will use only numbers placed in square brackets.

L51-52 -"[ 1](pp. 219-277), [2](p. 519), [3](pp.  200-204), [5](pp. 32-50),  [6](p. 38-50),  [7](pp.  82.-87), [8](pp.  11-13),  [9](pp.  37-5244), [10](pp. 365-370)" - See notes above.

This should be corrected throughout the text.

We have corrected those mistakes.

  • L109 -L110 " Many studies  also  show  that  the  percentage  of  inmates  who  smoke  tobacco,  drink alcohol or take drugs in a hazardous manner by far exceeds the corresponding proportion the general population. " - I can't see  quotes to such a bold thesis.

We have added the supporting quotes.

  • L123 - " social rehabilitation" - why italics was used?

We have corrected this mistake.

  • L125 - suggests adding a clear research aim.

Thank you for this suggestion. We added the research aim (L185-187).

Materials and Methods

  • L126-170 I miss more information about collecting data, combining it and conducting analyses. Who has consented to the analysis and access to the data. Have the inmates been informed that their data will be used? I suggest expanding this part.

We have added a detailed description about collecting data and analyzing process.

  • L127 - "This article is based on three different projects." I lack information about the ethical consent for each project or the written consent that the respondents have filled in to participate in it.

We have made the ethical side of the study described in a more detailed way. It is visible in the materials and methods section.

  • L145 - "through"- why italics was used?

We have corrected this mistake.

Results

  • L173-191 - This passage is more suited to the introduction than the results. You quote someone else's achievement, not your own.

We have rearranged this passage.

  • L177 - " [pl. Najwyższa Izba Kontroli]" change into "(pl. Najwyższa Izba Kontroli)

We have corrected this mistake.

  • L195 - "[IN THERAPY]" change into "[In Therapy]"

We have corrected this mistake.

  • L225 - "zero level" - why an underscore was used?

We have corrected this mistake which was a mistranslated slang expression.

  • L232 - "[NAME OF THE PRISON]" change into "[Name Of The Prison]"

We have corrected this mistake.

  • L356-L57 - "(BS6-BogusÅ‚aw / 60)" - I understand that "BogusÅ‚aw" is a name. I don't understand what this parenthesis means in the text. What was it placed for?

We have corrected this mistake, the parenthesis has been removed.

Discussion

  • L398 - change "[501" to "[50]"

We have corrected this mistake.

  • L421-422 - "Tony Ward and Shadd Maruna" - you should not use the first names of the authors in the text

We have corrected this mistake.

  • L451-459 - Should be used, initials used, not full names and surnames.

We have corrected this mistake.

  • Acknowledgments

L470 -"243"

L471 - "244"

L473 - "246" are these mistakes?

Yes, they were. We have deleted those numbers.

  • L473 -" Operational Programme Knowledge Education 246 Developmen" - add dot.

Dot added.

Once again we would like to sincerely thank you for a very comprehensive, insightful and in many places accurate review. All of the above comments are highly valuable for the comprehensiveness of the paper and our own scientific development.

Kind Regards,

Dariusz Timler Assoc. Prof., MD PhD

Reviewer 4 Report

Impersonal language should be used throughout the text (except in the discussion) and we and our should be avoided.
Reference 14 is from 2002, so it is very old and does not correspond to the country where the study is carried out.
71 sensory means sensory?
99 would put the quote in italics and in quotation marks
There is no prior reflective process for the research team.
It is good that three different studies have been done, but only the design is clear to me in the ethnographic one. What is the design of the other two and how has the data been analyzed beyond using Atlas.Ti?
Nothing is understood about the results, is it a thematic analysis and 3.1 and 3.2 are the themes? Within each of them there are many aspects that are not explained separately, even these results are debated when that should be done in the discussion.
It would be necessary to separate in some way the narratives of the interviewees, because what is narration is not distinguished from what is explanation.

Author Response

Dear Sir/Madam,

        We wish to express our appreciation for the comments and suggestions for our manuscript entitled “Risk factors of early adolescence in the criminal career of Polish offenders in the light of life course theory”. We have carefully revised the manuscript taking into consideration all the comments.

  • Impersonal language should be used throughout the text (except in the discussion) and we and our should be avoided.

We have corrected it throughout the text.

  • Reference 14 is from 2002, so it is very old and does not correspond to the country where the study is carried out.

We have switched this reference for the new one.

  • 71 sensory means sensory?

Yes. In other words we can talk about sensorial deprivation.

  • 99 would put the quote in italics and in quotation marks

We have rearranged the text and named paragraphs 3.1 and 3.2 in more clear way. We attempted to make the relation more apparent by strengthening the linkage between results and discussion.

We have changed the editorial style of the narratives of the interviewees so that it is more readable.

Once again we would like to sincerely thank you for a very comprehensive, insightful and in many places accurate review. All of the above comments are highly valuable for the comprehensiveness of the paper and our own scientific development.

Kind Regards,

Dariusz Timler Assoc. Prof., MD PhD

Round 2

Reviewer 1 Report

All my comments and suggestions have been addressed.

Reviewer 3 Report

I am impressed with the work you have done to improve the article. However, the article still has numerous editorial errors. I also have ethical objections which should be clarified.

General comments

1# ethical

L207-208 - "We also obtained general statistical data from the District Inspectorate of the Prison Service in Lodz, not violating medical confidentiality or accessing inmates’ personal data." Please indicate exactly what data was collected from these databases.

However, in my opinion, ,  even concerning this data, a consent should be given by participants.

" Participation by individuals capable of giving informed consent as subjects in medical research must be voluntary. Although it may be appropriate to consult family members or community leaders, no individual capable of giving informed consent may be enrolled in a research study unless he or she freely agrees."

WMA Declaration of Helsinki – Ethical Principles for Medical Research Involving Human Subjects

L213-226 - Were the subjects informed that they could withdraw their consent at any time? If so, add this information.

L213-226 - Did parents of juveniles agreed to their participation in their study? If yes, please eprovide their consent.

2# editorial error

My last general comment  - #6 has not been corrected. "6# References - incorrect description of many items. The authors should check all the items in the references description."

For example:

#8 - page numbers are missing

#10 - remove "No."

#12 - remove (12)

#15 -all the wrong form of quote.

#18 - page numbers are missing

The entire bibliography has to be corrected In this form it would not be accepted.

"Journal Articles:
1. Author 1, A.B.; Author 2, C.D. Title of the article. Abbreviated Journal Name Year, Volume, page range."

https://www.mdpi.com/journal/ijerph/instructions

If you use a bibliography manager (e. g. Zotero) use the mpdi template:

https://www.zotero.org/styles/?q=id%3Amultidisciplinary-digital-publishing-institute

Specific comments

L42-43: Key words: "social isolation/alienation",  please choose one option (for example social isolation or social alienation)

 L42-43: Key words: " and "health care/public health" - note above

L54-55 Change "[1 3,5-10]" to "[1,3,5-10]"

L155 - " (Annual Statistical Information for 2019/2020: 32)." what does that mean"32"?

L170 - " Diagram1. Number of inmates in the European Unioncountries (June 2021)" add period

L173 - change "(...)https://www.prisonstud-172ies.org/map/europe(June 23rd 2021). [43] " to "(...)https://www.prisonstud-172ies.org/map/europe(June 23rd 2021) [43]."

L174 - add period

L177 -change  " Authors’ own study based on data retrieved from: https://www.prisonstud-176ies.org/map/europe(June 23rd 2021). [43]" to " Authors’ own study based on dataretrieved from: https://www.prisonstud-176ies.org/map/europe(June 23rd 2021)[43]."

L182 - "(see Figure 3)" remove "see"

L187-189 – first add the quote and then the period.

L194 - change "[35.,47] " to "[35,47]".

L209-211 - "Moreover, the 209data collected from single facilities under theDistrict management were not suitable for statistical analyzes as they turned out to be rudimentary, undetailed and unstandardized." - I suggest to add this in the limitation section.

L217 -change  "project (in Polish) is available online [51].)." to "project (in Polish) is available online [51]."

L243 - change "No." to "no."

L739 - 930 -"References" - you use  different font and spacing

Reviewer 4 Report

Be aware doing a spoiler of the results (n participants) in methods